# Self-organized centripetal movement of corneal epithelium in the absence of external cues

Erwin P. Lobo[1], Naomi C. Delic[2,3], Alex Richardson[4], Vanisri Raviraj[2,3], Gary M. Halliday[2], Nick Di Girolamo[4], Mary R. Myerscough[1] & J Guy Lyons[2,3,5]

Maintaining the structure of the cornea is essential for high-quality vision. In adult mammals, corneal epithelial cells emanate from stem cells in the limbus, driven by an unknown mechanism towards the centre of the cornea as cohesive clonal groups. Here we use complementary mathematical and biological models to show that corneal epithelial cells can self-organize into a cohesive, centripetal growth pattern in the absence of external physiological cues. Three conditions are required: a circumferential location of stem cells, a limited number of cell divisions and mobility in response to population pressure. We have used these complementary models to provide explanations for the increased rate of centripetal migration caused by wounding and the potential for stem cell leakage to account for stable transplants derived from central corneal tissue, despite the predominantly limbal location of stem cells.

[1] School of Mathematics and Statistics, Camperdown, New South Wales 2006, Australia. [2] Discipline of Dermatology, Bosch Institute, Charles Perkins Centre, University of Sydney, Camperdown, New South Wales 2006, Australia. [3] Immune Imaging Program, Centenary Institute for Cancer Medicine and Cell Biology, Camperdown, New South Wales 2042, Australia. [4] Department of Pathology, School of Medical Sciences, University of New South Wales, Randwick, New South Wales 2052, Australia. [5] Sydney Head and Neck Cancer Institute, Cancer Services, Royal Prince Alfred Hospital, Camperdown, New South Wales 2050, Australia. Correspondence and requests for materials should be addressed to N.D.G. (email: n.digirolamo@unsw.edu.au) or to M.R.M. (email: mary.myerscough@sydney.edu.au) or to J.G.L. (email: g.lyons@centenary.org.au).

The cornea is the first part of the eye through which light must pass during the process of vision, and maintaining its clarity and geometrical structure is essential for high-quality vision in vertebrates. Corneal epithelial cells are derived largely from stem cells located in the limbus, a narrow collar of tissue that circumscribes the cornea[1–5]. Proliferation occurs exclusively within the basal layer of cells[6]. The population balance of corneal epithelial cells in adult eyes can be described by the X, Y, Z hypothesis, in which the proliferation and the migration of new epithelial cells into the cornea are counterbalanced by a loss of cells through terminal differentiation[7]. Corneal epithelial cells in adult mice under homeostatic conditions form spoke-like growth patterns[4,8,9]. By imaging living mice, we recently showed that these 'spokes' are clones of epithelial cells that stream continuously towards the centre of the cornea from the limbal margin, and account for the overwhelming majority of corneal epithelial cells in normal eyes[10]. This centripetal pattern of growth, together with observations of proliferative potential *in vivo* and *in vitro*, support the idea that the limbus is the main source of stem cells for the adult corneal epithelium[1,2,11]. These limbal epithelial stem cells (LESCs) are slow-cycling, long-lived and able to undergo both symmetric and asymmetric replication[3,5,12]. They give rise to transit amplifying cells (TACs), which have a limited proliferative capacity. LESCs

actively transcribe the keratin-14 gene (*KRT14*)[10,13,14]. Contrary to the generally accepted limbal location of stem cells[3,5,12], Majo and colleagues used tissue transplantation to demonstrate the existence of long-term epithelial progenitors in the central cornea[15], a finding that created considerable controversy[16].

The cause of the centripetal migration of corneal epithelium is not known, but several possibilities have been raised[12], including the production of chemotactic factors at the centre of the cornea, neuronal tracking, biochemical and biophysical differences in the underlying stroma, electrophysiological cues and population pressure differentials[17,18].

Here we show using a novel mathematical simulation model that corneal epithelial cells can self-organize into a cohesive, centripetal growth pattern in the absence of physiological cues from the rest of the cornea. We used the simulation model, together with analyses of mouse corneas, to show that ultraviolet radiation (UVR) accelerates centripetal migration by increasing apoptotic and non-apoptotic cell death. We also show that a low level of stem cell leakage from the limbus creates an accumulation of stem cells near the centre of the cornea, providing an explanation for previously described successful corneal transplants using limbus-deficient corneal tissue.

## Results

**Centripetal migration in a simulated cornea.** To address the mechanism of centripetal migration in the cornea, we developed a free-lattice mathematical simulation model (described in Methods and in the Overview, Design concepts, Details (ODD) protocol in Supplementary Methods). The cornea is represented as a circular collection of polygonal cells. Its rim forms a niche that supports LESCs, which have fixed locations (Fig. 1a). The

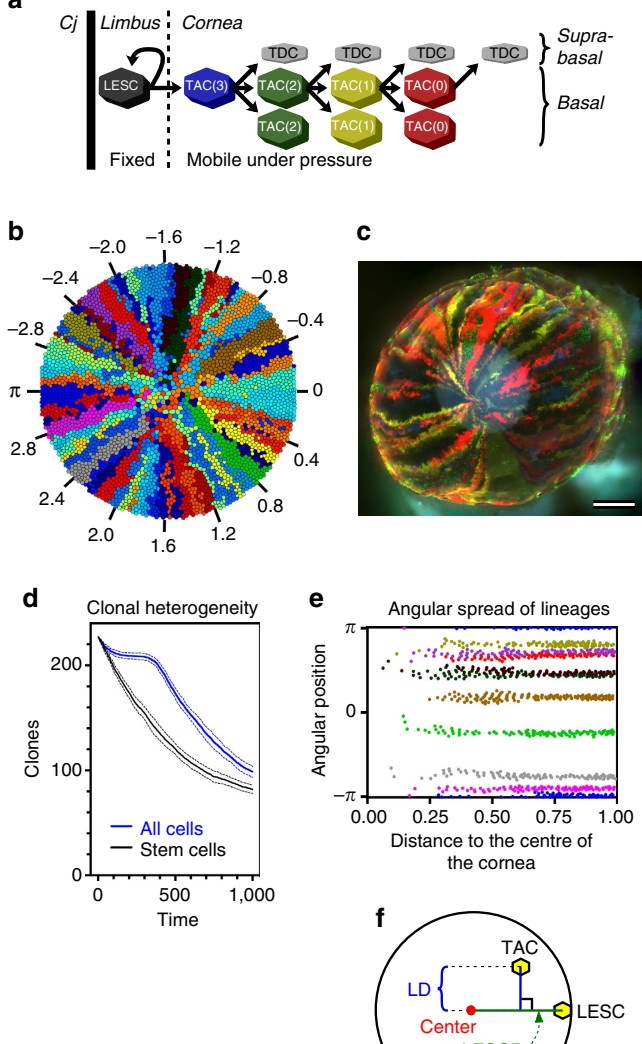

**Figure 1 | A mathematical model of the corneal epithelium recapitulates centripetal migration and clonal segmentation.** (**a**) A scheme of cell lineages in the cornea simulation model. The conjunctiva (Cj) forms an impenetrable perimeter. LESCs can divide either symmetrically, to produce two LESCs, or asymmetrically, to produce a LESC that is retained within the limbus and a TAC that occupies a space in the cornea immediately adjacent to the limbus. When a LESC dies, the two neighbouring LESCs are stimulated to replicate to fill the void, only one of them being able to occupy the space. TACs can divide either asymmetrically, to produce a suprabasal terminally differentiating cell (TDC) and another TAC in the basal layer, or symmetrically, to produce two TACs in the basal layer. TACs can undergo a limited number of symmetrical cell divisions, TAC(max), which is variable but typically set at 3, after which they die. The number of symmetric replications remaining for a TAC is shown in parentheses. (**b**) A representative simulation in which the replicative potential of TACs is limited to 4 symmetrical cell divisions and development has been allowed to continue for 750 iterations of the algorithm. The numbered spokes provide direction reference points (in radians) for the location of clones in **e**. (**c**) A fluorescence micrograph of a cornea from a K14CreERT2-Confetti mouse, injected with tamoxifen when 6 weeks old and killed 16 weeks later. The circular haze in the centre of the cornea is autofluorescence from the underlying lens. Scale bar, 500 μm. (**d**) The number of clonal lineages of all cells (blue) and only LESCs (black), plotted ± s.d. against time for 25 simulations. (**e**) The spread of cells of a lineage was plotted for 10 representative clones of the cornea shown in **b**. The distance is measured as a proportion of the corneal radius. The locations and mean linear displacements (LDs) of these clones are reported in Supplementary Table 1. (**f**) Calculation of the LD, a measure of the degree to which migration is centripetal. The locations of a TAC in a cornea and the LESC from which it is derived are shown. If migration of the LESC-derived lineage was perfectly centripetal, then the TAC would lie on the ray joining the position of the LESC to the centre of the cornea (the LESCR). The LD is the absolute distance of the perpendicular from the TAC to this ray.

lineage path that LESCs follow is essentially that described for corneal epithelia by Lehrer and colleagues[6]. Cells in the cornea exert a pressure on their immediate neighbours and, importantly, unlike LESCs, are motile, moving from high pressure towards low pressure. Total corneal pressure is kept constant by asymmetrical replication of LESCs, which compensates for TAC death. In the context of this model, death of a TAC includes apoptosis, division into two terminally differentiated cells, extrusion from the basal layer and any other process that ends the cell lineage. In our simulations only cells of the basal layer are depicted, because this is where cell reproduction occurs.

An example of the visual output of the model is depicted in Fig. 1b, which shows the clonal lineage map of a cornea after 750 time-steps of the simulation ($t = 750$). The colour of each LESC is inherited by TACs and other LESCs that derive from it. It can be seen that these conditions of peripheral stem cell location, limited replication capacity of TACs and movement in response to population pressure were sufficient to enable the cells to self-organize into spoke-like clonal lineage patterns, similar to those generated in the corneas of K14CreERT2-Confetti mice (Fig. 1c and refs 10,14). It was not necessary to invoke directed migration towards a source of a chemotactic factor or along predetermined paths to achieve this centripetal migration. This

spoke-like pattern was maintained in 1:1 scale representations of mouse corneas, that is, in simulations having a radius of 100 cells (Supplementary Fig. 1). In general, the cells of a clone were contiguous, but in some places neighbouring clones had invaded adjacent spokes, causing them to become discontiguous, a phenomenon that has also been observed *in vivo*[10,13] (Fig. 1b,c). From this example, it is also apparent that some clones within the cornea did not have a corresponding LESC of the same colour at this time-step. This occurred when all of the LESCs of a clone died and their space became occupied by neighbouring clones. This phenomenon of clonal extinction can be seen clearly in the video animation (Supplementary Movie 1) and led to a decrease in clonal heterogeneity of the cornea as it aged (Fig. 1d), as would be expected through neutral evolutionary drift[19,20]. The number of LESCs per surviving clone increased as the cornea aged, leading to a broadening and reduction in number of the spokes, similar to what is observed in mouse corneas[21]. This effect is analogous to the 'coarsening' of the clonal structure of a tissue generated from randomly dispersed stem cells[19].

To illustrate the cohesion of clones within the spokes more clearly, we plotted the angular location of a selection of clones (Fig. 1e). Perfect centripetal migration of a clonal lineage would

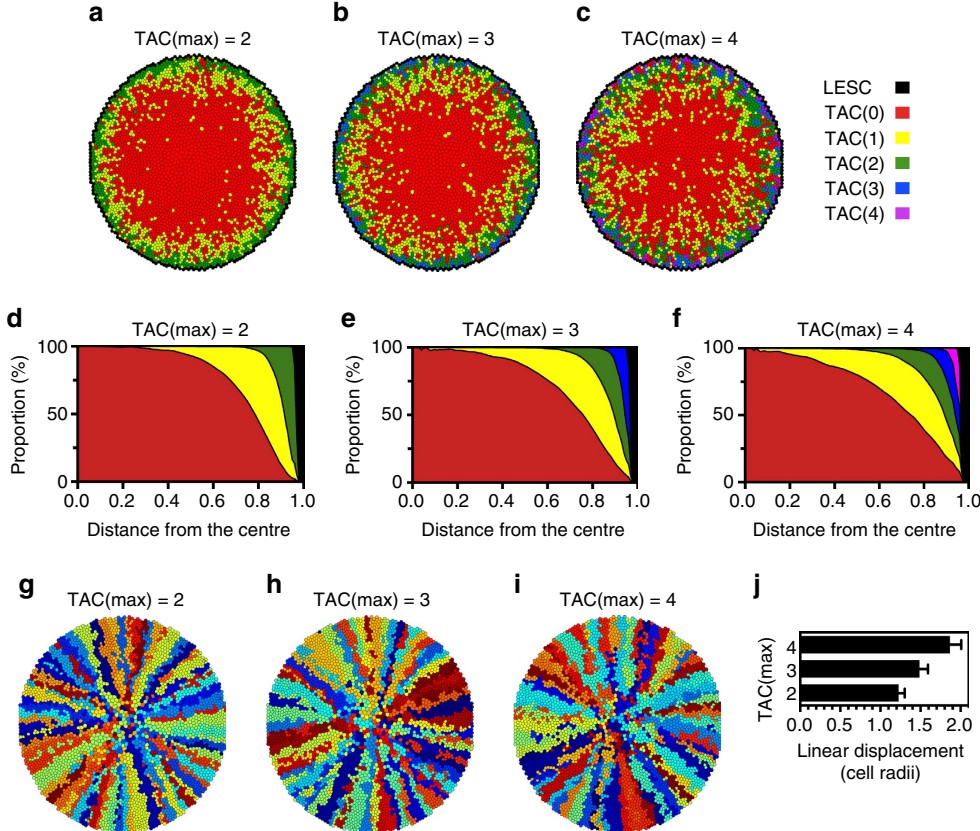

**Figure 2 | Centripetal migration of the corneal epithelium is retained with greater dispersal when replicative potential increases.** Representative simulations of corneal epithelium in which TAC(max), the maximal number of symmetric cell divisions in the basal layer after production of the TACs by the LESCs, was either 2 (**a,d,g**), 3 (**b,e,h**) or 4 (**c,f,i**). (**a–c**) Generation maps at $t = 1,000$ for each scenario. In each map, the LESCs are black cells circumscribing the cornea and the TACs that have reached their replicative limit are depicted in red. Cells with 1, 2, 3 or 4 symmetric cell divisions remaining are colour-coded according to the key at right. Cells throughout the cornea can give rise to suprabasal terminally differentiating cells, which are not shown. (**d–f**) Radial distribution of replicative potential within the corneal epithelium. Kernel density plots as a function of distance from the centre of the cornea are shown for each TAC(max) estimated from 25 simulations. The kernel density is an estimation of the underlying probability distribution for each cell category. The colours for the plots correspond to those in the generation maps in **a–c**. (**g–i**) The clonal lineage maps corresponding to the generation maps in **a–c**. (**j**) Increased spread of TACs away from ideal centripetal migration with increasing replication potential. The LDs were calculated for cells in corneas at $t = 750$ in which the TAC(max) was either 2, 3 or 4. The mean ± s.d. of the LDs are shown for 25 simulations of each case. The LDs of all three TAC(max) were different by one-way ANOVA with Tukey's multiple comparisons correction at a $P < 0.0001$.

result in the cells lying in a straight line, exactly following the ray connecting the LESC from which a clone was derived to the centre of the cornea (the LESCR, Fig. 1f). The spread of TACs away from the LESCR increased somewhat as the cells moved further from their LESC progenitor (Fig. 1e). To quantify the extent of centripetal migration, we calculated the mean linear displacement (MLD) of TACs, the linear displacement (LD) being the perpendicular distance of a TAC to the LESCR (Fig. 1f). The theoretical limits of the MLD are 0, which would indicate perfectly centripetal migration, and the length of the corneal radius, which would indicate active repulsion of the TACs away from the LESCR. Random dispersal of TACs would fall between these values. The MLD for these example clones, and the median

MLD of all clones, was very small ($<2$ cell radii, Supplementary Table 1), indicating a high level of cohesion of TACs to the LESCR, diagnostic of centripetal migration.

**Effect of clonal lifespan on centripetal migration.** The number of generations of TACs that a stem cell can give rise to, TAC (max), is a fundamental characteristic of epithelia that is limited by replicative senescence and telomere erosion[22]. Natural consequences of our population pressure-regulated model are that LESCs are slow-cycling compared with TACs, and the rate of LESC division decreases as TAC(max) increases (Supplementary Fig. 2). Thus, to maintain an equilibrium number of cells in the cornea, LESCs divided asymmetrically $3.16 \times 10^{-2}$ per cell per time-step when TAC(max) $= 2$, $1.48 \times 10^{-2}$ per cell per time-step when TAC(max) $= 3$ and $7.14 \times 10^{-3}$ per cell per time-step when TAC(max) $= 4$. Increasing the TAC(max) also caused the demarcation between regions dominated by cells of a generation to become less sharp, as can be appreciated visually in generation maps (Fig. 2a–c) and by the shallower slopes of the interfaces in the radial distribution graphs (Fig. 2d–f). Thus, when the initial replicative potential was high, cells of an earlier generation were more likely to be in the centre and cells of a later generation near the periphery than when the initial replicative potential was low. In other words, increasing the replicative potential resulted in corneas in which cells of different generations were more interspersed.

The spoke-like appearance of corneas was clearly maintained over a range of TAC(max) from 2 to 4 (Fig. 2g–i), which is the

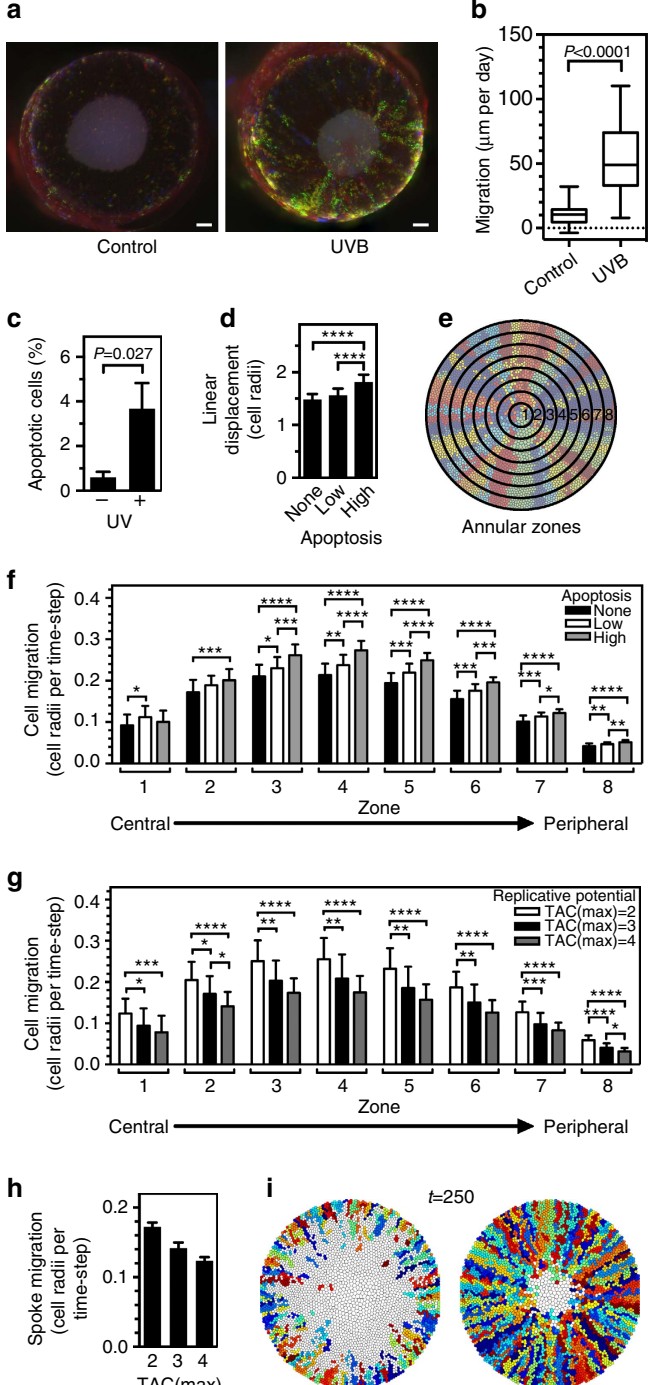

**Figure 3 | UVR damage drives centripetal migration of corneal epithelial cell clones.** (**a**) One eye of each K14CreERT2-Confetti mouse was irradiated with a single exposure of UVR ($150 \, \mathrm{mJ \, cm^{-2}}$). Eyes were obtained 7 days later and fluorescently imaged. The control (left) and irradiated eye (right) from a representative mouse are shown. Scale bar, $200 \, \mu\mathrm{m}$. (**b**) The lengths of contiguous streaks emanating from the limbal regions of both eyes of four mice were quantified. Box and whiskers represent the 25–75 percentile interval and the range, respectively. $P < 0.0001$ by Mann–Whitney $U$-test. (**c**) The proportion of basal epithelial cells undergoing apoptosis was determined by image analysis of confocal Z-stack micrographs of two regions of TUNEL- and DAPI-stained mouse corneas. The mean and s.e.m. of three mice are shown. The $P$ value is from a Student's $t$-test. (**d**) Increased spread of TACs away from ideal centripetal migration in wounded corneas. A wound was simulated by introducing a low (0.67%) or high (1.67%) rate of apoptotic cell death at each time-step throughout the cornea for 25 time-steps ($t = 651$–675). The LDs were calculated after wounding, for $t = 751$–760. The mean LD $\pm$ s.d. is shown for 25 simulations of each case. ****$P < 0.0001$ by one-way ANOVA with Tukey's multiple comparisons test. (**e**) Identification of the concentric annular zones used for analyses in **f**, **g** and elsewhere. (**f**) Corneal epithelial cell migration is enhanced during wounding. The mean radial cell migration rate was calculated during the wounding period ($t = 666$–675) in corneas wounded as in **d**, for each corneal annular zone, as shown in **e**. The mean and s.d. of 25 simulations are shown. *$P < 0.05$, **$P < 0.01$, ***$P < 0.001$, ****$P < 0.0001$ by one-way ANOVA with Tukey's multiple comparisons test. (**g**) Reducing the replicative potential of TACs increases cell migration. The mean radial cell migration rates in corneas in which TAC(max) was 2, 3 and 4 were calculated for each zone. The mean and s.d. of 25 simulations are shown. *$P < 0.05$, **$P < 0.01$, ***$P < 0.001$, ****$P < 0.0001$ by one-way ANOVA with Tukey's multiple comparisons test. (**h**) The net spoke migration rate was determined for 25 simulations in which TAC(max) was set at either 2, 3 or 4. All migration rates were significantly different from the others with $P < 0.0001$ by one-way ANOVA with Tukey's multiple comparisons test. (**i**) Representative images of simulated corneas at $t = 250$, demonstrating the impact that reducing TAC(max) has on increasing spoke migration, analogous to the response of real mouse corneas to UVR in **a**.

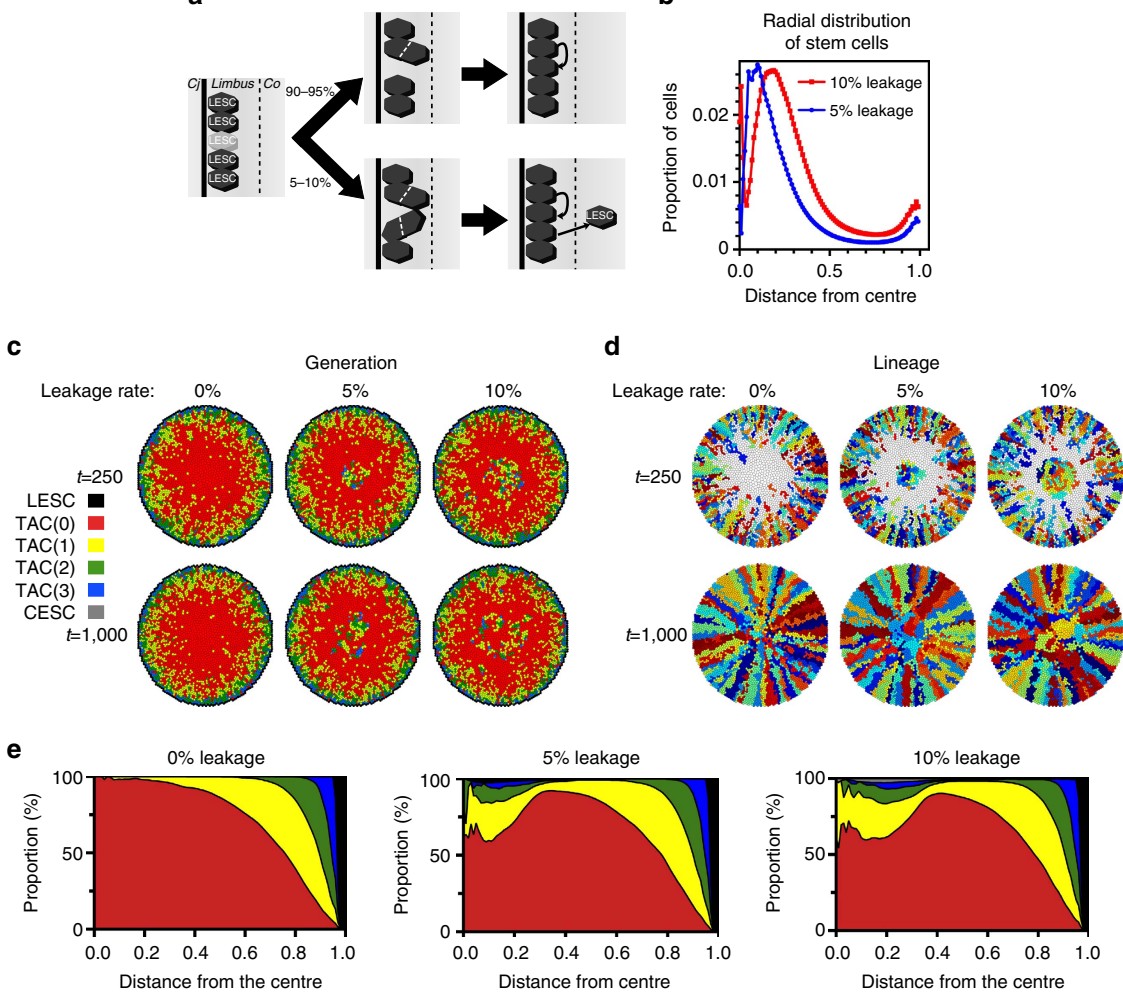

**Figure 4 | Stem cells leaking from the limbus accumulate near the centre of the cornea and reduce the degree of centripetal organization of the corneal epithelium.** (**a**) A scenario that might result in stem cell leakage from the limbus. The death of a LESC (left) stimulates neighbouring LESCs to replace it. Usually, only one neighbour successfully completes symmetrical cell division, leading to a one-for-one replacement (top). However, 5 or 10% of the time both neighbours respond synchronously, producing two stem cells to replace the dead cell (bottom). The excess stem cell is pushed into the cornea, becoming a CESC. Cj = conjunctiva; Co = cornea. (**b**) Bimodal distribution of stem cells resulting from their leakage from the limbus. The kernel density distribution plots are shown for stem cells in corneas undergoing 5% (blue) or 10% (red) leakage from the limbus. (**c**) The generation maps of three corneas, representative of 0, 5 and 10% stem cell leakage at $t = 250$ and $t = 1,000$, showing the accumulation of a small number of CESCs near the central cornea. (**d**) The clonal lineage maps corresponding to the generation maps in **c**. (**e**) Radial distribution plots of corneas at $t = 1,000$ undergoing 0, 5 and 10% stem cell leakage. Mean proportions of each cell generation at each distance from the centre of the cornea are shown ($n = 25$ simulations).

likely number of replications that they can undergo *in vivo*[6]. However, cells of corneas with a lower starting replicative potential showed more clonal cohesion than cells of a higher replicative potential, demonstrated by a significantly higher MLD when TAC(max) was 4 than when it was set at 2 (Fig. 2j). Overall, a lower TAC(max) gave rise to better organized corneas than a higher TAC(max) but demanded more frequent cycling of LESCs to maintain population equilibrium.

**UVR damage of the corneal epithelium.** To examine the effect of tissue damage on the behaviour of corneal epithelium, we wounded mouse corneas by exposing them to UVR, a natural cornea-damaging agent that can cause photokeratitis, pterygia and ocular cancers[23,24]. A single, low dose (150 mJ cm$^{-2}$) of broad-spectrum UVR (see Supplementary Fig. 3a for the spectrum) caused a fivefold increase in centripetal spoke migration (Fig. 3a,b). Staining whole-mounts of our corneas by

terminal deoxynucleotidyl transferase dUTP nick end labelling (TUNEL) showed evidence of apoptosis affecting an additional 3% of the basal cells 24 h after UVR exposure (Fig. 3c, Supplementary Fig. 3b). Previous investigations reported that apoptosis in response to UVR occurs over a 2–3-day period, peaking 1 day after exposure and returning to basal levels by 3 days[23].

To simulate a wound analogous to apoptotic death in corneas in the mathematical model we introduced an added probability of death of 0.67% (low damage) or 1.67% (high damage) per time-step to the TACs. High damage caused increased disorder within the corneas, evident from the overall MLD that was higher than that of a lower degree of wounding (0.67%) and controls (Fig. 3d). The immediate migratory response to wounding varied in different zones of the cornea. At the end of the wounding period, cell migration was enhanced in the intermediate zones (zones 2–6, Fig. 3e,f) but only slightly in the outermost zone. However, this effect did not last and, < 100 time-steps after wounding,

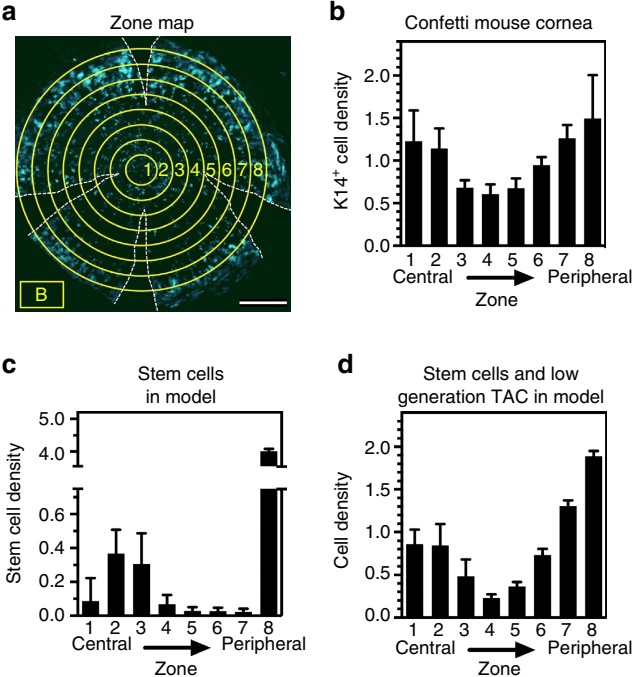

**Figure 5 | The distribution of stem cells in a mouse corneal lineage tracing model is consistent with stem cell leakage from the limbus.** The fluorescence images of corneas of K14CreERT2-Confetti mice (**a,b**) or simulation images (**c,d**) were divided into concentric zones and the density of the labelled cells was determined in each zone by calculating fluorescence intensity using ImageJ software. The cell density for each zone was normalized to the cell density of the entire cornea. (**a**) Confocal micrograph of a flat-mount of a representative cornea from an 8-week-old K14CreERT2-Confetti mouse, showing only the cyan channel. The mouse had been induced to undergo recombination 2 weeks earlier by the injection of tamoxifen. The gaps in the cornea created by the cuts made to flatten it for imaging are indicated by dotted lines. The concentric zones used to quantify the distribution of labelled cells (numbered 1–8) and the region used to adjust for background fluorescence (B) are indicated in yellow. Scale bar, 500 μm. (**b**) Quantification of the density of labelled cells per unit area within the different zones of the mouse cornea shown in **a**. A bimodal distribution of the cells, with peak densities at the periphery and near the centre, is evident. The mean ± s.d. of the distributions of the four colour channels (cyan, green, yellow and red) are shown. (**c**) Distribution of stem cells (combining limbal and corneal stem cells) in the simulated corneas at $t = 1,000$ with a stem cell leakage rate of 10%. The mean and s.d. of 10 simulations is shown. (**d**) The same simulated corneas as in **c** were re-analysed to account for low-generation TACs that would be labelled during the 2 weeks post tamoxifen in lineage tracing corneas such as that used in **a** and **b**. The data shown are the mean ± s.d. of the sum of stem cells and TAC generations 1–3.

migration returned to similar rates as control corneas (Supplementary Fig. 3c). Thus, the biological data in conjunction with the simulations suggested that an alternative, non-apoptotic response would be needed for enhanced migration of corneal epithelium to be sustained beyond 2–3 days of the apoptosis induced by UVR. One such alternative response could be a shortened generational lifespan of clones, denoted by a reduced TAC(max). Simulations showed that, in contrast to a UVR response that increased apoptosis, a response that reduced the TAC(max) would cause a significant increase in individual cell migration throughout the cornea (compare the migration rates in zones 1, 7 and 8 of both low and high apoptosis in Fig. 3f with TAC(max) = 2 in Fig. 3g). This reduced TAC(max) caused an

increase in the spoke migration rate like that seen *in vivo* (Fig. 3h,i). Thus, apoptosis contributes to the increased migration of corneal epithelium during the first 2–3 days after wounding by UVR. Beyond that, a longer-lasting non-apoptotic response that continues to reduce the generational lifespan of clones can account for the increased migration rate.

**Stem cell leakage from the limbus.** Next, we used our simulation model to identify a mechanism that could reconcile the generally accepted notion that stem cells are located predominantly in the limbus with the findings of Majo and colleagues[15], in which the central cornea was capable of restoring corneal integrity when transplanted to the limbus. We postulated that LESCs might undergo occasional cell divisions that are symmetric in phenotype (that is, producing two daughter stem cells), but asymmetric in location (that is, one retained in the limbus and the other pushed into the cornea), and still give rise to the centripetal migratory patterns seen *in vivo*. For example, if the death of a LESC were a stimulus for its two neighbouring LESCs to undergo symmetric replication, on some occasions both LESCs might respond synchronously (Fig. 4a). In that scenario, two additional LESCs would be produced but there would be room in the limbus for only one of them, the other being evicted into the cornea, becoming a corneal epithelial stem cell (CESC). This would create a leakage of stem cells from the limbus into the cornea. Alternatively, the leakage might occur stochastically.

We simulated scenarios in which stem cell leakage happened 5 or 10% of the time that LESCs replicated (Fig. 4; Supplementary Fig. 4). This low level of stem cell leakage resulted in a bimodal distribution, with stem cells accumulating both near the centre and at the periphery of the cornea (Fig. 4b,c,e, Supplementary Movies 2–4). This bimodal distribution of stem cells (Fig. 4b) formed a less ordered pattern of clones, particularly near the centre of the cornea (Fig. 4c,d), resulting in a MLD that was significantly higher in the central regions than in corneas with no stem cell leakage (Supplementary Fig. 5). Nevertheless, the clones retained a clearly recognizable spoke-like arrangement (Fig. 4d) with very low MLDs: 1.48, 1.56 and 1.56 cell radii for stem cell leakage rates of 0%, 5% and 10%, respectively. Age and genetic differences might affect the rate of stem cell leakage, and genes such as *NOTCH1* (which was knocked out in some of the mice used by Majo and colleagues[15]) could affect the lifespan of these stem cells[25]. Indeed, we observed that this central accumulation of CESCs was dependent on a relatively long life for the stem cells; reducing their lifespan from 10 times to twice that of TACs virtually abolished their accumulation in the centre. We next used the model to simulate the transplantation of CESCs to the centre of a cornea in the absence of stem cell leakage from the limbus. These CESC-derived cell lineages did not form spoke-like lineage patterns and were not stable, being outcompeted by lineages derived from LESCs (Supplementary Fig. 6).

To determine whether the stem cell leakage model is consistent with *in vivo* observations, we analysed their distribution in corneas from the K14CreERT2-Confetti lineage tracing mice[10] that had been injected with tamoxifen 2 weeks previously, giving rise to small clusters of labelled cells encompassing the stem cells (Fig. 5a). There is a major peak of stem cells in the peripheral zone near the limbus and a smaller peak near the centre (Fig. 5b). The relative distribution of cells in the central and outer corneal zones varied among different animals, but the bimodality was consistent (Supplementary Fig. 7). By applying the same methodology used for the mouse corneas to the generation maps produced by our simulation model, a similar bimodal distribution of stem cells can be seen. In this case, the stem cell population is dominated by a major peak at the limbus, but about

5% is located in a second peak near the centre (zones 1–4, Fig. 5c). When we included not just stem cells, but also TACs with a generation of three or fewer, to better reflect the generation status in the mouse corneas 2 weeks post tamoxifen, a relatively higher proportion of cells accumulated near the centre, similar to that seen *in vivo* (Fig. 5d).

## Discussion

Our simulation model shows that centripetal migration in the corneal epithelium can be explained through population pressure forces, without the need to invoke signals mediated by soluble morphogens, extracellular matrix molecules, neurons or biophysical cues. However, it does not rule out a role for those signals in mediating proliferative and migratory responses to conditions such as chronic inflammation, wound healing and intraocular pressure[2,6,15,21,26–28]. During homeostasis, the location of the stem cells within the limbus, together with the limited replicative capacity of TACs and the mobility of epithelial cells in response to pressure, is sufficient to produce both centripetal migration and cohesive self-organization of clonal lineages into sectors. The centripetal nature of the migration is robust, being maintained to a high degree over a range of TAC(max) (Fig. 2), wounding (Fig. 3), stem cell leakage (Fig. 4), the maximum number of movement steps to allow pressure equilibrium to occur (Supplementary Fig. 8a) and TAC lifespan (Supplementary Fig. 8b). The result is remarkably similar in appearance to the coloured spoke-like patterns that develop in K14CreERT2-Confetti corneas[10,14]. The loss of TACs when they pass their TAC(max) generation naturally creates a population pressure drop in the middle of the cornea relative to the periphery, because the cells near LESCs are more likely to be younger and therefore die at a lower rate. The simplicity of this mechanism is appealing as one that might have evolved as a means of efficiently maintaining the homeostasis of a tissue that is important to the survival of many organisms. The location of stem cells at the periphery of the cornea would offer them some protection from UVR in organisms in which the eyelids cover part of the limbus or in which melanocytes are present in the limbus.

Exposure to agents such as UVR that increase cell death also cause an increase in the rate of centripetal migration. Conformity to the X, Y, Z hypothesis demands that an enhancement of cell death is needed to compensate for the increased rate of influx of TACs into the cornea. Thus, to sustain a faster centripetal migration beyond the reported 2–3 days over which apoptosis is reported to occur[23], either a high rate of apoptosis would need to be sustained for longer or an alternative mechanism of accelerated cell death would need to follow it. The enhanced migration and maintenance of centripetal pattern seen in UVR damaged corneas is consistent with there being a signal to reduce the TAC(max). This would both increase the migration rate (Fig. 3) and reduce the cycling time of the LESCs in damaged corneas as has been observed both in the simulation model (Supplementary Fig. 2) and in mouse corneas that have been chemically wounded[6]. The molecular mechanism that could reduce the TAC(max) is unknown, but could conceivably involve an increase in telomere erosion and/or extrusion of TACs from the basal layer.

The simulation model also shows that stem cell leakage from the limbus is a simple explanation for the presence of stem cells in the central corneal epithelium, while acknowledging their predominantly limbal location. It is conceivable that, even under conditions of low stem cell leakage, there would be enough stem cells in the central cornea to replace LESCs if they were transplanted to the edge of a cornea from which the limbus had been removed[15]. Manipulating the rate of stem cell leakage might have therapeutic value for conditions in which the cornea is damaged or deficient in these cells, by analogy with hemopoietic stem cells.

## Methods

**Mathematical model.** We created a free-lattice, discrete state mathematical model in Matlab to represent the basal layer of the cornea, based on geometrical structures known as 'Voronoi diagrams'. The model is described in detail in Supplementary Methods according to the Overview, Design concepts, Details (ODD) protocol[29]. Cells in the cornea exert a pressure on their immediate neighbours and move from high pressure towards low pressure according to Equation 1 (Supplementary Methods). At each time-step, the positions of the cells are adjusted by applying this movement rule algorithm to equalize pressure locally, thereby filling 'holes' that are created by the death of TACs. The repetition of the algorithm ceases when all of the cells in the cornea are less than 2 cell diameters away from all their neighbouring cells or, failing this, after 100 applications of the algorithm. Total corneal pressure is kept constant by asymmetrical replication of LESCs, which compensates for TAC death. Death includes any process that ends the cell lineage. Supplementary Table 2 provides a comparison of key parameters observed in mouse corneas and those that were either chosen as input or arose as output from the simulation model. For computational tractability, most simulations were done on a cornea of 4,000 cells, which is about one-third of the diameter of a real mouse eye. Unless otherwise stated, the following standard conditions were used for simulations: an average lifespan of TACs = 75 time-steps; a replicative potential of TAC(max) = 3; no leakage of stem cells from the limbus; a maximum pressure equilibration process of 100 iterations per time-step.

Unless otherwise stated, the MLDs in simulations were calculated at $t = 750$ and the migration rates of cells were determined by tracking every cell in the cornea that was present at $t = 750$ through to $t = 760$, calculating the net radial distance travelled and dividing by 10.

Net spoke migration rates of simulations were calculated for each of the six 60 degree sectors of corneas from the number of frames taken for the first cell to reach the inner edge of zone 3. This was conditional on that cell having a continuous chain of non-white cells to the perimeter, so as to avoid counting isolated cells that were not part of a spoke; isolated cells analogous to these would not have been counted in mouse corneas when determining migration rates of entire spokes.

**Mouse lineage tracing and wounding model.** K14CreERT2-Confetti mice[30,31] were used and imaged as described previously[10]. The use of mice was approved by the University of Sydney Animal Ethics Committee under protocol 0604 and University of NSW protocol 14/89B. Six-week-old female and male K14CreERT2-Confetti mice were injected intraperitoneally with $25 \, mg \, kg^{-1}$ of tamoxifen (Sigma T5648) in 10% ethanol, 90% sunflower seed oil (Sigma S5007) on two consecutive days. The mice were then anaesthetized with ketamine/medetomidine and both corneas of each mouse were imaged using a Nikon AZ100 wide-field fluorescent microscope with CoolLED4000 light source, to obtain a baseline measurement. While sedated, the right eye of each mouse was protected from UVR with aluminium foil to serve as a control, and the left eye was exposed to $150 \, mJ \, cm^{-2}$ UVR from 2 Oliphant FL40SE fluorescent tubes (Oliphant-UV, Adelaide, SA, Australia) with the addition of a cellulose triacetate filter to omit UVC wavelengths. Randomization of mice was not necessary because experimental and control eyes were obtained from all mice. The ultraviolet irradiance spectrum was determined using an Optronics OL-754 spectroradiometer[24] and found to be ~40% UVB + 60%UVA. After 7 days, the corneas were fluorescently imaged again to determine the increase in spoke length. Four spokes per eye from each channel (cyan, green, yellow and red) were randomly chosen and measured. Their corresponding spokes from day 0 were also measured, and the difference between before and after UVR was corrected to a measurement of µm per day.

**K14$^{+}$ corneal cell distribution.** K14CreERT2-Confetti mice were injected with tamoxifen, as above, and 2 weeks later the fluorescence in their corneas was imaged. In some animals, Evans blue dye was injected intravenously to clearly define the limbus by highlighting its vasculature in the red fluorescence channel. For *ex vivo* imaging, all four channels (cyan, green, yellow and red) were analysed on whole mounts. The images were divided into eight concentric annuli whose intensities of fluorescence were measured using Fiji software. For eyes imaged *in vivo*, the fluorescence images of corneas were also divided into sectors and only those sectors in which the entire radial distribution from centre to limbus was present were analysed, using the yellow channel.

**Corneal cell death analysis.** Mice used for apoptosis studies were Cre negative Confetti mice. Following euthanasia, mouse eyes were removed, fixed in 4% paraformaldehyde in phosphate-buffered saline and then the corneas were dissected away from the rest of the eyes. Cells undergoing apoptosis in mouse corneas were identified using a DeadEnd Fluorometric TUNEL staining kit (Promega). Stained corneas were counterstained using 4′,6-diamidino-2-phenylindole (DAPI)($0.1 \, µg \, ml^{-1}$) and then imaged using a Zeiss LSM 510 confocal microscope.

The proportion of apoptotic cells was calculated from Z-stacks as the number of TUNEL positive cells divided by the number of basal epithelial cells. For each eye, one field from the centre and one field from the intermediate region (approximately zones 3–6) were analysed.

**Data availability.** The authors declare that all relevant data and computer code supporting the findings of this study are included in the manuscript and/or available on request from the corresponding authors.

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

## Acknowledgements

This work was supported by the Human Frontiers Science Program Grant RGP0041-2011 to J.G. Lyons and M.R. Myerscough and Australian National Health and Medical Research Council grant 1101078 to N. Di Girolamo. N.C. Delic and A. Richardson are supported by Australian Postgraduate Awards and N.C. Delic is also supported by a Sydney Catalyst Top-Up Scholarship from Cancer Institute NSW. We acknowledge the assistance of Louise Cole and Cathy Payne of the Bosch Institute Advanced Microscopy Facility, University of Sydney, and Dr Iveta Slapetova of the UNSW Biomedical Imaging Facility. Thanks to Professor Wolfgang Weninger for critical reading and comments.

## Author contributions

E.L. was the programmer and principal designer of the simulation model and performed the *in silico* experiments; J.G.L. conceived the project; N.C.D., A.R. and V.R. performed *in vivo* experiments; all authors assisted with design of the program and experiments, interpreted the results and wrote the paper.

## Additional information

**Competing financial interests:** The authors declare no competing financial interests.

**How to cite this article**: Lobo, E.P. *et al.* Self-organized centripetal movement of corneal epithelium in the absence of external cues. *Nat. Commun.* **7:**12388 doi: 10.1038/ncomms12388 (2016).

