## [Peer Review File · Nature Communications]

Reviewer #1 (Remarks to the Author)

A. Summary of the key results. The stem cells that regenerate the cornea are believed to reside in the limbus, a narrow collar of cells between the cornea and the sclera. In this paper, the authors produce a computational model of corneal cell replacement to confirm that the limbal position of epithelial stem cells, coupled with pressure-driven motility and cell turnover, can produce the spoke-like pattern often observed in lineage-tracing experiments. There was controversy in the field when limbus-free transplants successfully regenerated corneal epithelia. However, the authors show that leakage of stem cells from the limbus creates a subpopulation of motile stem cells in the interior of the cornea, thus explaining the surprising experimental results.

B. Originality and interest. This paper builds a model of corneal cell replacement using two constraints: that the corneal stem cells reside in the limbus and that population pressure differentials govern cell motility. While the mathematical modeling techniques are not novel, nor the idea that the stem cells reside in the limbus, they were able to show that these two mechanisms are sufficient to form the spoke-like lineage patterns observed in the cornea.

The most novel idea in this paper is the idea that leakage of stem cells from the limbus can cause the interior of the cornea to be studded with transient populations of stem cells. This insight unifies the widely held theory that the limbus contains the stem cells of the cornea with surprising experimental observations that central corneal transplants can also regenerate the epithelia. However, more simulations, especially in the context of wounding, need to be done to show that this mechanism is sufficient to account for the experimental findings.

C. Data and methodology.

The update rules governing the model are very intuitive and easy to understand. The quality and presentation of the data is excellent, although some of the figures could be improved by condensing the information. For example, in Fig1e/f, it may be better to plot the distributions of the angular spread, number of cells generated, mean resultant length, etc. Also, the authors only need to show one representative trace for the angular spread of lineages. Ideally these values should also be measured from images of the K14CreERT2-Confetti corneas, to show how the simulated and experimental data compare.

D. Statistics

Statistical tests are sound.

E. Conclusions

The conclusions summarize the findings well. However, the discussion on the putative mechanistic reasons for why the ESCs reside in the limbus, while interesting, doesn't serve the function of supporting the findings. Ideally the conclusions should either restate the argument in support of the findings, or discuss the implications of the findings.

F. Suggested improvements.

Other work in the field has shown the pattern of regeneration given wounding or trauma. Create a wounding model in an eye with an intact limbus to see whether clonal clusters emerge in the pattern, as discovered in ref 12. Also, create a model to test central cornea transplantation into a limbus-free eye, to

see whether the patterns of regeneration match the experimental findings.

The positioning of the LESC within the limbus is not clearly explained. Fig 1a gives the impression that each LESC is isolated within the limbus. However, Fig 3a shows each LESC as being surrounded by other LESC, which is also implied in the supplementary information. If this is indeed the case, then it seems unintuitive that each LESC can generate clones that occupy such a large radial slice of the corneal surface. In a related note, in Fig1d, it's unclear why the number of clones decrease over time if the the limbus is populated entirely by LESC, which are anchored there. To address this, it may be very useful to zoom in on the limbus and show a timecourse of the cellular dynamics near the boundary.

It's hard to understand the take-home message of the simulations in Figure 2, because they seems unrelated to the main point about stem cell organization in the cornea. It would be more interesting if there were a value of TAC(max) that completely breaks the model, but it seems like the highest values they tested still create spoke-like patterns. It would also be interesting to directly compare these values to experimental estimates of TAC lifetimes, if these have been determined. If neither of these two lines of inquiry yield interesting conclusions, it may be better to move Figure 2 and related discussion to the supplemental and condense it to a smaller point.

Can the authors define a metric for centripetal arrangement, and plot this metric across a variety of test cases?

Besides the spoke-like pattern, are there archetypal patterns that can be formed by alternative arrangements of stem cells within the cornea? What would a fixed disk in the interior of the cornea look like? Or a fixed ring in the interior of the cornea. How does limbal cell deficiency manifest in the pattern?

The cornea is multiple layers thick, but the computational model is only a single cell layer thick. Are the LESC arranged in a single layer or are they distributed along the z-axis of the cornea at the limbus? If they are in a single layer, how does the layered architecture affect cellular motility and patterning?

Please articulate the physical basis of the cell movement rule.

A table of parameter values (as well as the rationale for their values) would help in deciphering the validity of the model.

G. Previous work. Previous work is appropriately referenced

H. Clarity and context. The abstract and presentation of data can be further sharpened. The main idea is that the dynamic geometrical model of the cornea provides a general framework with which to test different theories about corneal epithelial cell dynamics during homeostasis and regeneration. However, in the body of the work, not a lot of effort is devoted to testing a diverse set of potential models. Mainly, it would be interesting to see what cell structures are formed with scattered and concentrated corneal epithelial stem cells, as well as what cellular structures are formed during wound healing processes.

One main result is that passive cell mobility based on population pressure differentials and the placement of stem cells in the limbus is enough to generate the spoke-like pattern that has been consistently observed in the cornea. Given this, they should also foreground the simulation of other models (i.e. that of corneal epithelial stem cells (CESCs) dominating homeostasis) to show that they do not generate the correct lineage geometries. Although the model given in S2 shows that CESCs eventually disappear over time, it would be interesting to see what geometries are produced if those CESCs are positionally anchored (just

like the limbus cells) either in a central disc or scattered throughout the epithelium. Figure 4 would be enhanced by graphs showing what the stem cell density would look like given other architectures with stem cell leakage.

Another idea they test using their model is that leakage of stem cells from the limbus can produce pockets of stem cells within the middle of the cornea. They say that this could explain why central cornea transplantations can regenerate the cornea, but they still need to further buttress this point by running simulations of regeneration after wounding, with direct comparisons to experimental results from the literature.

Reviewer #2 (Remarks to the Author)

The authors develop a discrete cell based model to understand the stem cell maintenance of the corneal epithelium. The authors propose a simple model in which transit amplifying cells (derived from limbal stem cells) divide a limited number of times and experience population-pressure induced movement, and show that this is sufficient to explain the centripetal movement of cells from the periphery to the centre. Further, allowing symmetric differentiation of stem cells created a leakage of limbal stem cells that pool at the centre of the cornea, consistent with some recent findings. Overall, this is an impressive paper which I am favourable towards and I would be happy to see it published. My comments focus on the modelling side of the paper: I do not have the necessary technical expertise to comment on any experimental work.

(1) The size of the "model eye" (~4000 cells) would seem relatively small compared to a real eye. I appreciate that this is due to the demands of the model, in that simulating a real-sized eye may be computationally intractable at present. However, perhaps some comment on this issue (and the extent it could impact on results) could be in order.

(2) The authors provide a fairly detailed model description, although I'm not 100% sure how easy it would be to reproduce their exact methods: this is of course a typical and understandable problem with discrete cell models. However, perhaps some statistics related to how much the population "shifts" in a single time step? The manuscript offers a method based on repeatedly shifting cell positions, however I found it hard to judge how much this would shift (on average) a single cell in a single time step, and therefore allow some more quantitative comparison between the model time step and movement rates in vivo (if known). On this issue, how sensitive are the results to the "computational parameters" (for example the exact rules that determine when the movement algorithm stops)? I guess may major question here is the extent to which such aspects were simply "plucked from the air" or more rigorously chosen?

Reviewer #3 (Remarks to the Author)

In this manuscript the authors use a mathematical techniques to model and confirm the hypothesis that corneal epithelial stem cells (SCs) are preferentially located in the limbal epithelium and their progeny (TACs) populate the corneal epithelium via centripetal migration. This manuscript is well-written and referenced appropriately. As mentioned, this work is in many ways confirmatory rather than novel.

Comments:

1. The authors must consider the idea that, in reality, the early progeny (TACs) of a SC may be phenotypically, kinetically and biochemically indistinguishable from a SC (an idea originally proposed by Potten in the 90's). They need to acknowledge this and build it into their model.

2. The author's model predicts that a low level of stem cell leakage creates an accumulation of stem cells near the center of the cornea. This is inconsistent with the available biological data. For example, one of the expected ways of detecting stem cells is the formation of holoclone colonies. As shown by Majo and others, the formation of holoclone colonies derived from corneal cells is species dependent. Most importantly holoclone colony formation does not occur in the human. Additionally, the presence of label-retaining cells is another well-accepted technique for localizing stem cells and such label-retaining cells have not been detected in the corneal epithelium. Finally, population doublings (a measure of proliferative potential, as shown by Pelligrini et al), which is another hallmark of stem cells is high in the limbus and dramatically lower in the peripheral and central cornea. Therefore, the authors must revise and or deal with lack of bonafide SCs in the cornea in their model.

3. It is not clear what the model predicts during wound healing and or tear gland insufficiency (dry eye). The authors need to consider these conditions as they are some of the most frequently encountered clinical problems.

4. In Figure 4a, do the other fluorescent colors show the same distribution like the cyan channel?

Response to Reviewers' Comments

Reviewer 1

1. In Fig1e/f, it may be better to plot the distributions of the angular spread, number of cells generated, mean resultant length, etc. ... the authors only need to show one representative trace for the angular spread of lineages. Ideally these values should also be measured from images of the K14CreERT2-Confetti corneas

We have transferred the data in Fig. 1f into Supplementary Material as Supplementary Table 1, as requested. We have retained the existing Fig. 1e, because it most clearly conveys the generality of the tightness of angular dispersion of the clones, which a single trace could not. Showing just 1 trace would not save much space compared to showing 10. Measuring these distributions in K14CreERT2-Confetti mouse corneas is not feasible, as it would require knowledge of the exact location of the stem cell in the limbus that gave rise to each TAC, which is not known for these samples.

2. The discussion on the putative mechanistic reasons for why the ESCs reside in the limbus, while interesting, doesn't serve the function of supporting the findings. Ideally the conclusions should either restate the argument in support of the findings, or discuss the implications of the findings.

We have removed the discussion relating to the location of blood vessels, as requested. However, we have retained the discussion point relating to protection from ultraviolet radiation (UVR) as a possible evolutionary driver of locating stem cells at the periphery of the cornea, because we have now included findings of the effects of UVR in the manuscript. Therefore, it has become an implication of the finding.

3. Create a wounding model in an eye with an intact limbus to see whether clonal clusters emerge in the pattern, as discovered in ref 12. Also, create a model to test central cornea transplantation into a limbus-free eye, to see whether the patterns of regeneration match the experimental findings.

Both Reviewer 1 and Reviewer 3 requested to see the simulation model applied to the situation in which a cornea is damaged, such as the DMSO damage model of reference 12. To address this, we have undertaken experiments using an *in vivo* experimental model in which the cornea is damaged by UVR and a complementary simulation model. These experiments demonstrate that an increase in cell death rate in the corneal epithelium is sufficient to greatly increase the rate of centripetal migration, which is seen in UVR treated K14CreERT2-Confetti corneas in living mice. UVR occurs naturally as a component of sunlight and causes photokeratitis ("snow blindness"), pterygia and ocular cancers and so is a more realistic source of damage than DMSO. The data from this mouse model of corneal damage and the corresponding mathematical simulations are described in Fig. 3 and lines 135-160 on pages 6 and 7.

4. It seems unintuitive that each LESC can generate clones that occupy such a large radial slice of the corneal surface. In a related note, in Fig1d, it's unclear why the number of clones decrease over time if the limbus is populated entirely by LESCs, which are anchored there. To address this, it may be very useful to zoom in on the limbus and show a timecourse of the cellular dynamics near the boundary.

In the model, the LESCs have a lifespan that is much longer than that of TACs, but finite nevertheless. Thus, individual LESCs die at random times. When they do so, the progeny of one of the 2 neighbouring LESCs occupies the space vacated by the dead LESC, thereby increasing the radial slice occupied by the replacement clone. When the LESCs of a clone are completely replaced by neighbouring clones, that clone becomes extinct and the number of clones in that cornea decreases by one. This phenomenon has been observed previously in the mouse cornea¹ and

other epithelial tissues² and represents a spatially constrained example of an evolutionary process known as "neutral drift"^{2,3}. We have made this clearer in the text by rewording the text and including references on page 4, lines 87-96.

5. It's hard to understand the take-home message of the simulations in Figure 2, because they seems unrelated to the main point about stem cell organization in the cornea. It may be better to move Figure 2 and related discussion to the supplemental and condense it to a smaller point

A limited number of cell divisions is one of the 3 basic conditions that we identified that are needed to maintain centripetal growth of corneal epithelium, and is the only 1 of the 3 conditions that can be varied quantitatively. Thus, the maximum number of cell divisions that TACs can undergo [TAC(max)] is a fundamental characteristic of the simulation model that deserves some examination. The data in Figure 2 are important because they demonstrate that the replicative potential of TACS, as well as determining the cycling rate of LSCs, affects the degree of centripetal organization of the cornea. In this revised version of the manuscript, a change in the replicative potential is also identified as a mechanism that could drive the enhanced rate of centripetal migration of clones seen in UVR-treated mouse corneas. The other significance of the simulations in Figure 2 is that they demonstrate the robustness of the model: centripetal migration is retained to a high degree for a range of TAC(max) that is consistent with biological observations. We have re-titled the Figure to clarify this message.

6. Can the authors define a metric for centripetal arrangement ... ?

The mean resultant length (MRL) in the original manuscript measured how directly a cell reached its destination in the cornea from the ESC from which it originated. This has been replaced in the revised manuscript by the mean linear displacement (MLD), which measures how close the cells within a clonal spoke are to the ray joining the ESC from which they originated to the centre of the cornea. The MLD, therefore, measures how centripetal the migration is, rather than just how direct it is. This is defined on page 5, lines 102-108 and in Figure 1f in the revised manuscript. Data throughout the original manuscript that reported MRL have been replaced in the revised manuscript with the corresponding MLD.

7. ... Plot this metric across a variety of test cases? ... Besides the spoke-like pattern, are there archetypal patterns that can be formed by alternative arrangements of stem cells within the cornea? What would a fixed disk in the interior of the cornea look like? Or a fixed ring in the interior of the cornea. How does limbal cell deficiency manifest in the pattern? ...

We agree that the simulation model will be useful in the future to explore various aspects of corneal responses to developmental and pathological stimuli, and possibly to the development of bioengineered tissues. However, for this communication we restricted our modelling of homeostasis to scenarios related to corneal maintenance for which there is biological evidence i.e. a location of stem cells in the limbus, which is almost universally accepted, and accumulation of leaked stem cells in the central cornea, for which we provide evidence. We also modelled in Supplementary Fig. 4 (now numbered Supplementary Fig. 6 in the revised manuscript) the scenario in which stem cells are transplanted to the centre of the cornea, as done by Majo and colleagues. Finally, in this revised manuscript we have now also included data from both simulations and *in vivo* experiments on corneas wounded by UVR (Fig. 3 and Supplementary Fig. 3). Although it might be of mathematical interest to carry out simulations with various arbitrary geometric starting points of the stem cell locations, in this paper it would be inappropriate and distract from the focus on modelling real corneas.

8. The cornea is multiple layers thick, but the computational model is only a single cell layer thick. Are the LSCs arranged in a single layer or are they distributed along the z-axis of the cornea at

the limbus? If they are in a single layer, how does the layered architecture affect cellular motility and patterning?

Although the cornea is multiple layers thick, the evidence in the literature overwhelmingly supports the idea that cell proliferation occurs in the basal layer; this applies both to stem cells in the limbus and to transit amplifying cells in the cornea proper^{4,5}. This is now stated explicitly in the Introduction, with a supporting reference (page 2, line 30), in addition to page 3, lines 71-73. The simulation model focuses on cell lineages and the proliferation that gives rise to them. Although suprabasal layers could be incorporated into the model, they would not add information about corneal epithelial lineage proliferation and population maintenance. It would also make visualisation of the cells less clear by superimposing layers over one another.

9. Please articulate the physical basis of the cell movement rule.

The cells move purely in response to local population pressure gradients created by the loss of cells through death or gain of cells through replication. We have expanded upon the plain English description of the model in Methods to clarify the cell movement rule with the following text (page 11, lines 258-265): " Cells in the cornea exert a pressure on their immediate neighbors and move from high pressure towards low pressure according to Equation 1 (ODD, Supplementary Methods). At each time-step, the positions of the cells are adjusted by applying this movement rule algorithm in order to equalize pressure locally, thereby filling "holes" that are created by the death of TACs. The repetition of the algorithm ceases when all of the cells in the cornea are less than 2 cell diameters away from all their neighbouring cells or, failing this, after 100 applications of the algorithm." A detailed technical description of movement rule is also included in the ODD Protocol in Supplementary Methods.

10. A table of parameter values (as well as the rationale for their values) would help in deciphering the validity of the model.

Supplementary Table 2 is included in the revised manuscript and contains parameters that were used to inform the model or arise as an outcome of the model.

11. The main idea is that the dynamic geometrical model of the cornea provides a general framework with which to test different theories about corneal epithelial cell dynamics during homeostasis and regeneration. However, in the body of the work, not a lot of effort is devoted to testing a diverse set of potential models. Mainly, it would be interesting to see what cell structures are formed with scattered and concentrated corneal epithelial stem cells, as well as what cellular structures are formed during wound healing processes.

Please see response to Reviewer 1, point 7.

12. They should also foreground the simulation of other models (i.e. that of corneal epithelial stem cells (CESCs) dominating homeostasis) to show that they do not generate the correct lineage geometries.

Please see response to Reviewer 1, point 7.

Reviewer 2

1. The size of the "model eye" (~4000 cells) would seem relatively small compared to a real eye. I appreciate that this is due to the demands of the model, in that simulating a real-sized eye may be computationally intractable at present. However, perhaps some comment on this issue (and

the extent it could impact on results) could be in order.

The diameter of the model cornea is about 1/3 of the diameter of a real mouse cornea, in terms of cell numbers. This is now stated explicitly in Methods on page 11, lines 269-271, as well as in the ODD protocol in Supplementary Methods. As the reviewer noted, running all of the simulations as a 1:1 scale "map" would be computationally intractable. Nevertheless, in order to show that the basic principle of centripetal migration extends to a cornea of 200 cells diameter, the size of a mouse cornea⁶, an example has been included as Supplementary Fig. 1.

2. Perhaps some statistics related to how much the population "shifts" in a single time step? The manuscript offers a method based on repeatedly shifting cell positions, however I found it hard to judge how much this would shift (on average) a single cell in a single time step, and therefore allow some more quantitative comparison between the model time step and movement rates in vivo (if known)

We have calculated the net spoke migration rate for real mouse corneas and simulated corneas and included the values in Fig. 3b and 3h, respectively. When converted to equivalent units, they show good correspondence i.e. 1.4 cell radii per day for real corneas and 1.2 cell radii per day for simulated corneas (Supplementary Table 2).

3. How sensitive are the results to the "computational parameters" (for example the exact rules that determine when the movement algorithm stops)? I guess may major question here is the extent to which such aspects were simply "plucked from the air" or more rigorously chosen?

We have conducted simulations in which several parameters were varied. In the revised manuscript, these include the degree of wounding (Fig. 3), the time spent in pressure equilibration in the movement rule (Supplementary Fig. 8a) and TAC lifespan (Supplementary Fig. 8b), as well as the variations in TAC(max) (Fig. 2) and stem cell leakage (Fig. 4, Supplementary Fig. 5) that were originally included. The centripetal migration is remarkably robust, being largely maintained under all conditions and locations within the cornea, despite being somewhat perturbed by increasing the TAC(max) and the degree of wounding of the rate of stem cell leakage, as noted in the text. The pressure equilibration time for the movement algorithm and the TAC lifespan have little effect. This is now explicitly stated in Discussion on page 9, lines 218-222.

Reviewer 3

1. This work is in many ways confirmatory rather than novel

We disagree. The novelty of this work lies not in identifying the location of the stem cells in the limbus, which we agree is firmly established in the literature, but rather in the following discoveries: (1) a mechanism and identification of the minimal requirements for centripetal migration, which surprisingly do not include morphogenic gradients or other anatomical factors; (2) stem cell leakage from the limbus as an explanation for the ability to transplant tissue from the central cornea to replace the stem cell population in the limbus. In addition, in our revised submission we have included novel biological and computational data on UVR wounding of the cornea and its effects at a cellular and tissue level.

2. The authors must consider the idea that, in reality, the early progeny (TACs) of a SC may be phenotypically, kinetically and biochemically indistinguishable from a SC (an idea originally proposed by Potten in the 90's). They need to acknowledge this and build it into their model.

Potten and Loeffler proposed the idea of epithelial "potential stem cells", which are early TACs that can revert to a stem cell phenotype if they are able to occupy a vacated position in the stem cell niche⁷. Given that they are indistinguishable from stem cells in so many respects, evidence for their existence *in vivo* is, understandably, lacking, and so including them in our model would be speculative. It would also be of dubious value for the following reasons:

- Centripetal migration is generated in our model without needing to invoke the existence of "potential stem cells".
- If "potential stem cells" do happen to exist in the cornea, then the generation maps in Fig. 2a-c and Fig. 4c make it clear that, being early generation TACs, they would be adjacent to the limbus, not present in the central cornea.
- The centripetal flow of cells away from the stem cell niche in the limbus would make it unlikely that such a cell would occupy a vacancy created by the death of a LESC. Moreover, such an event, if it were to occur, would have no effect on the clonal lineage pattern of the corneal epithelium, because the "potential stem cell" would replace the stem cell from which it had been immediately derived.

3. The author's model predicts that a low level of stem cell leakage creates an accumulation of stem cells near the center of the cornea. This is inconsistent with the available biological data.

We disagree that our model is not supported by available biological data. Our model places the majority of stem cells in the limbus and a minority in the central cornea. Our own data in Fig. 5a and Supplementary Fig. 7 demonstrate that stem cells, marked by keratin-14 gene transcriptional activity *in vivo*, follow the distribution suggested by stem cell leakage.

As detailed below, there is also evidence in the literature that supports the presence of stem cells in the cornea. In other papers, where there is an absence of evidence for stem cells in the cornea, the methods used might not have been sensitive enough to detect them. The proportion of stem cells that is located in the central cornea in our model is less than 5% of the total, even when there is a 10% rate of leakage of stem cells from the limbus (Fig. 5c). This number of stem cells might be sufficient to repopulate a cornea with epithelial cells through clonal expansion, as described in the paper of Majo *et al.*⁸, but difficult to detect when using some of the methods mentioned by Reviewer 3 that have been used previously to identify stem cells (see also the response to point 4, below).

In addition to the Majo *et al.* paper, there are several credible supporting reports in the literature that support the presence of stem cells within the central cornea. For example:

- Dua *et al.* (2009) identified a number of patients with Limbal Stem Cell Deficiency who presented with persistent zones of healthy central corneal epithelium⁹.
- In an organ culture model, upon removing the human limbal collar, central corneal epithelia migrate centrifugally (i.e. towards the limbus) to repair the defect¹⁰.
- Others have shown that rabbit limbal epithelium does not participate in healing central corneal wounds even after several consecutive rounds of debridements^{11,12} and that ablating murine limbal epithelial stem cells then cauterizing the central cornea, resulted in normal re-epithelialization¹³.

Thus, the observations that support the presence of central corneal stem cells and contravene the firmly entrenched dogma of an exclusively limbal location of stem cells need to be consolidated into a model that acknowledges a predominantly limbal location of stem cells. For this reason we have proposed the "stem cell leakage" model, which is consistent with all of these data.

4. For example, one of the expected ways of detecting stem cells is the formation of holoclone colonies. As shown by Majo and others, the formation of holoclone colonies derived from corneal cells is species dependent. Most importantly holoclone colony formation does not occur in the human. ... Finally, population doublings (a measure of proliferative potential, as shown by Pelligrini et al), which is another hallmark of stem cells is high in the limbus and dramatically lower in the peripheral and central cornea. Therefore, the authors must revise and or deal with lack of bonafide SCs in the cornea in their model.

We agree with this Reviewer that the formation of holoclone colonies and population doubling assays *in vitro* has been an accepted strategy for estimating stem cell content and stem cell activity, particularly in the absence of *in vivo* lineage tracing models¹⁴⁻¹⁶. Cell culture amplification has also been a useful tool for the enrichment of stem cells for practical applications¹⁷. However, these *in vitro* assays are very indirect measurements of stem cell numbers *in vivo* and have limitations in their ability to quantify and compare stem cells derived from different tissues. They rely on the stem cells being efficiently isolated from the tissues and retaining their stem cell properties in cell culture, which provides a very different microenvironment from that of either the limbus or the cornea. Using these assays quantitatively to compare the frequency of stem cells in 2 different tissues, such as cornea and limbus, would need to assume an equal yield of stem cells from the 2 tissues and an equal ability to grow in culture, despite these tissues having a very different constitution of cell types and extracellular matrix, or to take into account these inequalities of yield and growth potential. However, neither the efficiency of stem cell isolation from different tissues nor the sensitivity of holoclone formation as an assay of *in vivo* stem cells has been determined in the various studies in the literature. Our *in vivo* lineage tracing model, with its persistent fluorescent streaks that develop from keratin-14⁺ cells marking stem cell derived clones, provides a much more direct measure of stem cells than holoclone formation or population doubling *in vitro*.

5. Additionally, the presence of label-retaining cells is another well-accepted technique for localizing stem cells and such label-retaining cells have not been detected in the corneal epithelium.

We agree that the presence of label-retaining cells (LRC) is a well-accepted technique for locating stem cells. However, the sensitivity of the LRC-based assay can be low, making them hard to detect: Cotsarellis et al¹⁸ could identify "few if any" LRCs in the limbus in normal, unchallenged mice, despite the generally accepted notion that stem cells reside there; wounding or chemical treatment of the cornea was required to be able to detect LRCs in the limbus. Nevertheless, contrary to Reviewer 3's belief that these cells have not been detected in the corneal epithelium, evidence is provided in the following papers:

- Barbosa et al (2009) identified LRC in the central cornea after corneal epithelial debridement and wound resolution¹¹.
- Haddad (2000) identified rare LRC in the central cornea 3 months post-labelling and proposed that the proliferative capacity of centrally located cells was sufficient to guarantee corneal epithelial renewal under steady-state¹⁹.

6. It is not clear what the model predicts during wound healing and or tear gland insufficiency (dry eye). The authors need to consider these conditions as they are some of the most frequently encountered clinical problems.

Please see response to Reviewer 1, point 4.

7. In Figure 4a, do the other fluorescent colors show the same distribution like the cyan channel?

Yes, the channels all have similar distributions. We chose to show only the cyan channel because we felt that the distribution of cells was clearer when viewing only 1 channel. Fig. 4b from the original submission (now labelled Fig. 5b in the revised submission) incorporates the distribution of all 4 channels. An image showing all channels is included in this response, below, but we do not think that it would be beneficial to add it to the manuscript. Additionally, we include new data from other mouse eyes in Supplementary Fig. 7, which all show a bimodal distribution.

References cited in Response to Reviewers' Comments

1. Mort, R. L., Ramaesh, T., Kleinjan, D. A., Morley, S. D. & West, J. D. Mosaic analysis of stem cell function and wound healing in the mouse corneal epithelium. *BMC Dev. Biol.* **9**, 4 (2009).
2. Klein, A. M. & Simons, B. D. Universal patterns of stem cell fate in cycling adult tissues. *Development* **138**, 3103-3111 (2011).
3. Orr, H. A. Fitness and its role in evolutionary genetics. *Nat. Rev. Genet.* **10**, 531-539 (2009).
4. Lehrer, M. S., Sun, T. T. & Lavker, R. M. Strategies of epithelial repair: modulation of stem cell and transit amplifying cell proliferation. *J. Cell Sci.* **111**, 2867-2875 (1998).
5. Lavker, R. M., Tseng, S. C. & Sun, T. T. Corneal epithelial stem cells at the limbus: looking at some old problems from a new angle. *Exp. Eye Res.* **78**, 433-446 (2004).
6. Di Girolamo, N. *et al.* Tracing the fate of limbal epithelial progenitor cells in the murine cornea. *Stem Cells* **33**, 157-169 (2015).
7. Potten, C. S. & Loeffler, M. Stem cells: attributes, cycles, spirals, pitfalls and uncertainties. Lessons for and from the crypt. *Development* **110**, 1001-1020 (1990).
8. Majo, F., Rochat, A., Nicolas, M., Jaoude, G. A. & Barrandon, Y. Oligopotent stem cells are distributed throughout the mammalian ocular surface. *Nature* **456**, 250-254 (2008).
9. Dua, H. S., Miri, A., Alomar, T., Yeung, A. M. & Said, D. G. The role of limbal stem cells in corneal epithelial maintenance: testing the dogma. *Ophthalmology* **116**, 856-863 (2009).
10. Chang, C. Y., Green, C. R., McGhee, C. N. & Sherwin, T. Acute wound healing in the human central corneal epithelium appears to be independent of limbal stem cell influence. *Invest. Ophthalmol. Vis. Sci.* **49**, 5279-5286 (2008).
11. Barbosa, F. L., Goes, R. M., de Faria, E. S. S. J. & Haddad, A. Regeneration of the corneal epithelium after debridement of its central region: an autoradiographic study on rabbits. *Curr. Eye Res.* **34**, 636-645 (2009).
12. de Faria-e-Sousa, S. J., Barbosa, F. L. & Haddad, A. Autoradiographic study on the regenerative capability of the epithelium lining the center of the cornea after multiple debridements of its peripheral region. *Graefes Arch. Clin. Exp. Ophthalmol.* **248**, 1137-1144 (2010).
13. Vauclair, S. *et al.* Corneal epithelial cell fate is maintained during repair by Notch1 signaling via the regulation of vitamin A metabolism. *Dev Cell* **13**, 242-253 (2007).
14. Barrandon, Y. & Green, H. Three clonal types of keratinocyte with different capacities for multiplication. *Proc. Natl Acad. Sci. USA* **84**, 2302-2306 (1987).
15. Pellegrini, G. *et al.* Location and clonal analysis of stem cells and their differentiated progeny in the human ocular surface. *J. Cell Biol.* **145**, 769-782 (1999).
16. Di Iorio, E. *et al.* Isoforms of DeltaNp63 and the migration of ocular limbal cells in human corneal regeneration. *Proc. Natl Acad. Sci. USA* **102**, 9523-9528 (2005).
17. Corradini, F., Venturi, B., Pellegrini, G. & De Luca, M. Methods for characterization/manipulation of human corneal stem cells and their applications in regenerative medicine. *Methods Mol Biol* **916**, 357-372 (2012).
18. Cotsarelis, G., Cheng, S. Z., Dong, G., Sun, T. T. & Lavker, R. M. Existence of slow-cycling limbal epithelial basal cells that can be preferentially stimulated to proliferate: implications on epithelial stem cells. *Cell* **57**, 201-209 (1989).
19. Haddad, A. Renewal of the rabbit corneal epithelium as investigated by autoradiography after intravitreal injection of 3H-thymidine. *Cornea* **19**, 378-383 (2000).

Reviewer #1 (Remarks to the Author)

The authors have done a great job in this revision, both by revising the text and figures to more clearly reflect their main points, and also by experimentally validating a wounding model whose behavior is predicted by the computational model. I would recommend this paper for publication, but the authors should consider revising (or just simplifying) their presentation of the wounding model for maximum clarity.

Specific comments:

Revision of the abstract is well done and effectively communicates the main findings of the research.

The explanation of how LESC population dynamics at the limbus affects the radial spoke patterns of the cellular clones is clearer now.

Figure 2: TAC(0) vs TAC(4) is unintuitive number labeling. It's easier for me to think of the number in the parentheses as the number of divisions that have already occurred rather than the number of divisions remaining. If the authors think this is generally true for most people in an audience, it may be clearer to change the labeling.

The wounding model is very convincing in showing that changing the dynamics of apoptosis (effectively TAC(max)) can drive centripetal migration of clones. However, figures 3f-g are somewhat confusing. From rereading the discussion of those results and looking at the graphs 3 times, it became clear that the authors are trying to communicate that the UVR wounding response does not exert its effects solely from increasing apoptotic rate, but rather that it likely reduces the max lifetime of a TAC. This could be clarified in the presentation.

Figures 4 and 5 communicate the stem cell leakage model and experimental validation very well.

The definition of the mean linear displacement is clearer and easier to follow.

Reviewer #2 (Remarks to the Author)

I have read through the revised manuscript. The authors have provided a careful and thorough response to the points I raised in my previous review, and have produced a revised version that I find acceptable for publication. I would, however, stress that I am not an expert on the biological aspects, and do not comment on that component of the manuscript.

Reviewer #3 (Remarks to the Author)

Response to Reviewers' Comments on the Revised Manuscript

Reviewer 1

The authors have done a great job in this revision, both by revising the text and figures to more clearly reflect their main points, and also by experimentally validating a wounding model whose behavior is predicted by the computational model. I would recommend this paper for publication, but the authors should consider revising (or just simplifying) their presentation of the wounding model for maximum clarity.

Specific comments:

Revision of the abstract is well done and effectively communicates the main findings of the research. The explanation of how LESC population dynamics at the limbus affects the radial spoke patterns of the cellular clones is clearer now.

Figure 2: TAC(0) vs TAC(4) is unintuitive number labeling. It's easier for me to think of the number in the parentheses as the number of divisions that have already occurred rather than the number of divisions remaining. If the authors think this is generally true for most people in an audience, it may be clearer to change the labeling.

We understand the Reviewer's point on the numbering system and gave this a lot of consideration before deciding to use the number of remaining divisions to label the cells, rather than the number of divisions that had already occurred. There are two reasons for this:

(1) Conceptually, we have found that it is cell death, rather than cell division, that provides the pressure gradient that drives centripetal migration. Thus, a count-down to the death of a clone seems to be more useful to monitor than a tally of the number of cell divisions that have occurred.

(2) From a practical viewpoint, by using the number of remaining divisions we can use consistent color-coding for the generation maps, such as those in Fig. 2a-c. In contrast, if we were to use the number of divisions that had already occurred to number the cells, then the cells that can no longer divide – which always comprise close to half of the total cell number and form most of the cells near the center – would be colored differently in Fig. 2a, b and c, making visual comparison difficult.

Therefore, we feel that it is best to retain the generational numbering as it is.

The wounding model is very convincing in showing that changing the dynamics of apoptosis (effectively TAC(max)) can drive centripetal migration of clones. However, figures 3f-g are somewhat confusing. From rereading the discussion of those results and looking at the graphs 3 times, it became clear that the authors are trying to communicate that the UVR wounding response does not exert its effects solely from increasing apoptotic rate, but rather that it likely reduces the max lifetime of a TAC. This could be clarified in the presentation.

To clarify and emphasize the implications of Figs 3f and 3g and Supplementary Fig. 3c, which were correctly interpreted by the Reviewer, we have revised the explanatory text in lines 157-167. However, we have retained Figs 3f and 3g in their current form. These figures compare the effects of apoptosis and non-apoptotic reduction of TAC(max) on migration rate. If we were to simplify them by pooling the data from different zones, they would lose the information on the spatial distinctions within and between the two scenarios.

Figures 4 and 5 communicate the stem cell leakage model and experimental validation very well. The definition of the mean linear displacement is clearer and easier to follow.

Reviewer 2

I have read through the revised manuscript. The authors have provided a careful and thorough response to the points I raised in my previous review, and have produced a revised version that I find acceptable for publication. I would, however, stress that I am not an expert on the biological aspects,

and do not comment on that component of the manuscript.

Reviewer 3

Reviewer #3 choose to leave remarks to the Editor only.